# Development of an Ultrasonic Doppler Sensor-Based Swallowing Monitoring and Assessment System

**DOI:** 10.3390/s20164529

**Published:** 2020-08-13

**Authors:** Younggeun Choi, Minjae Kim, Baekhee Lee, Xiaopeng Yang, Jinwon Kim, Dohoon Kwon, Sang-Eok Lee, HyangHee Kim, Seok In Nam, Saewon Hong, Giltae Yang, Duk L. Na, Heecheon You

**Affiliations:** 1Department of Industrial and Management Engineering, Pohang University of Science and Technology, Pohang 37673, Korea; sidek@postech.ac.kr (Y.C.); minjae@postech.ac.kr (M.K.); kjinwon30@postech.ac.kr (J.K.); kydson@postech.ac.kr (D.K.); 2Body Test Team 3, Hyundai Motor Company, Hwaseong 18280, Korea; x200won@hyundai.com; 3School of Artificial Intelligence and Computer Science, Jiangnan University, Wuxi 214122, China; yxp233@jiangnan.edu.cn; 4Department of Rehabilitation Medicine, Pohang Stroke and Spine Hospital, Pohang 37659, Korea; neopyte75@hotmail.com; 5Graduate Program in Speech-Language Pathology, Department and Research Institute of Rehabilitation Medicine, Yonsei University College of Medicine, Seoul 03722, Korea; h.kim@yonsei.ac.kr; 6Graduate School of Social Welfare, Yonsei University, Seoul 03722, Korea; namseokin@yonsei.ac.kr; 7R&D Research Team, Digital Echo Co., Hwaseong 18385, Korea; swhong@digital-echo.co.kr; 8R&D Research Team, SEED Tech. Co., Bucheon 14523, Korea; seele@seedtech.co.kr; 9Department of Neurology, Samsung Medical Center, Sungkyunkwan University School of Medicine, Seoul 06351, Korea; dukna@naver.com

**Keywords:** dysphagia, swallowing, assessment, ultrasonic Doppler sensor, quantification, correlation

## Abstract

Existing swallowing evaluation methods using X-ray or endoscopy are qualitative. The present study develops a swallowing monitoring and assessment system (SMAS) that is nonintrusive and quantitative. The SMAS comprises an ultrasonic Doppler sensor array, a microphone, and an inertial measurement unit to measure ultrasound signals originating only from swallowing activities. Ultrasound measurements were collected for combinations of two viscosity conditions (water and yogurt) and two volume conditions (3 mL and 9 mL) from 24 healthy participants (14 males and 10 females; age = 30.5 ± 7.6 years) with no history of swallowing disorders and were quantified for 1st peak amplitude, 2nd peak amplitude, peak-to-peak (PP) time interval, duration, energy, and proportion of two or more peaks. The peak amplitudes and energy significantly decreased by viscosity and the PP time interval and duration increased by volume. The correlation between the time measures were higher (*r* = 0.78) than that of the amplitude measures (*r* = 0.30), and the energy highly correlated with the 1st peak amplitude (*r* = 0.86). The proportion of two or more peaks varied from 76.8% to 87.9% by viscosity and volume. Further research is needed to examine the concurrent validity and generalizability of the ultrasonic Doppler sensor-based SMAS.

## 1. Introduction

The early screening and intervention of dysphagia are of importance because dysphagia can cause aspiration, pneumonia, dehydration, and malnutrition, resulting in a decreased quality of life and even death [1]. Dysphagia, a disturbance in transferring solid or liquid food from the mouth to the stomach [2,3], occurs in neurologic patients and the elderly [4,5]. Note that laryngeal closure and vocal fold closure occur involuntarily during pharyngeal swallowing to prevent food from entering the airways [6,7,8,9]. Examples of dysphagia prevalence statistics include 16.1% of adults aged 18 years or above (2353 males and 2645 females; mean age = 46.5 ± 15.7 years) in the USA [10], 35.3% of frail elderly people (29 males and 56 females; age = 80.8 ± 7.6 years) in Japan [4], 27.2% of older adults aged 70 years or above (136 males and 118 females; age = 78.2 ± 5.6 years) in Spain [11], and 23.6% of older adults aged 65 years or above (195 males and 220 females; age = 76.5 ± 8.6 years) in the Republic of Korea [12]. Ebihara et al. [13] and Sasaki [14] reported that aspiration pneumonia is a leading cause of death after cancer, heart diseases, and cerebrovascular diseases among older adults aged 65 years or above.

The videofluoroscopic swallowing study (VFSS) and the fiberoptic endoscopic evaluation of swallowing (FEES) are commonly used to diagnose dysphagia despite their limitations of visual inspection-based assessment. VFSS is a video-based X-ray examination of an entire swallowing process to diagnose the functional disorder of swallowing. However, repetitive VFSS can cause severe tissue injury to the patient by the over-exposure of X-rays [15]. Furthermore, VFSS relies mainly on a qualitative evaluation of clinicians and requires a significant amount of time for an objective assessment to measure the displacement of the hyoid-larynx complex or the transit time of the bolus using image analysis software [16,17,18,19,20]. FEES is an endoscopic assessment method which examines velopharyngeal closure, laryngeal elevation, vocal cord movement, pyriform sinus congestion, oropharyngeal function, penetration, aspiration, and presence of residue [21,22,23,24,25,26,27]. However, FEES also relies mainly on a qualitative evaluation of clinicians and can cause discomfort to the patient by the insertion of the nasopharyngoscope into the naris along the floor of the nose to the pharynx. Furthermore, FEES cannot directly visualize the aspiration nor assess the oral and esophageal phases of swallowing [28]. Thus, a dysphagia assessment system is needed to overcome the limitations of VFSS and FEES in terms of objective evaluation.

Acoustic methods that record swallowing sounds to evaluate the pharyngeal swallowing function have the limitation of capturing signals which can be contaminated due to extraneous sources other than swallowing activities. Previous studies [29,30,31,32,33,34,35,36] have conducted acoustic analyses on swallowing sounds using a stethoscope, microphone, or ultrasonic Doppler sensor, and examined the effects of gender, age, the viscosity of bolus, and the volume of bolus on the peak intensity and duration of swallowing signals. The acoustic methods need to provide a module to remove noises due to respiration, vocalization, coughing, neck movement, and/or the rubbing motion of the microphone against the skin from their measurements.

The present study develops a swallowing monitoring and assessment system (SMAS) consisting of a strip-type ultrasonic Doppler sensor array, a microphone, and an inertial measurement unit (IMU) and examines the characteristics of ultrasonic Doppler signals by pharyngeal swallowing of healthy participants. Ultrasound measurements only due to pharyngeal movements during swallowing are extracted by referring to signals of the microphone and IMU that monitor vocalization, coughing, and other body motions. Lastly, quantitative measures of ultrasonic Doppler signals are proposed and then their characteristics including the effects of the viscosity and volume of liquid bolus on the ultrasound measures and the relationships between the ultrasound measures are explored.

## 2. Development of Swallowing Monitoring and Assessment System

A wireless neck-band-type SMAS (see Figure 1a) has been developed by continuously upgrading the previous models developed by our research team [37,38,39,40,41,42], comprising an ultrasonic Doppler sensor array, a microphone, an IMU, and a Bluetooth module. The ultrasonic sensor array placed on the neck records signals caused by movements of the internal structures of the neck at the pharyngeal area for swallowing, vocalization, coughing, and other motions around the neck, while the microphone and the IMU placed on the chin record those caused by vocalization, coughing, and other motions around the neck. Thus, ultrasound signals are excluded as noises from swallowing analysis if values of the microphone or IMU are greater than designated values (e.g., microphone >0.2 mV or IMU >200 deg/s). A seven-ultrasonic Doppler sensor array with two transmitters and five receivers in a curved strip (Figure 1b; DA-2.0M7-CRBK, Digital Echo Co., Republic of Korea; frequency = 2 MHz ± 3%, length (L) × width (W) × thickness (T) = 38.0 mm × 15.5 mm × 3.0 mm, kerf = 1 mm, and the radius of curvature = 133 mm) is custom-designed for its close contact to the neck and securing a wide capture area of reflected ultrasound signals due to a pharyngeal movement. The sensor array is housed in a strip pad to measure movements in a sufficiently large pharyngeal area and connected to the device by a flexible polyvinyl chloride (PVC) cable. Next, an omnidirectional electret condenser microphone (WM-54BH, Panasonic Corporation, Japan; frequency = 20~16,000 Hz, radius = 9.7 mm, sensitivity = −42 ± 2 dB, and signal to noise ratio >60 dB) and an IMU (MPU-6500, TDK Corporation, Japan; 6-axis gyro accelerometer, L × W × T = 3 mm × 3 mm × 0.9 mm) are used to record signals due to vocalization, coughing, and/or movements around the neck, which also can affect ultrasound signals. Lastly, a Bluetooth chip (nRF52832, Nordic Semiconductor, Norway; frequency = 2.4 GHz and data rate = 2 Mbps) is used for wireless communication of signals.

A signal processing program is developed in the present study for the acquisition, rectification, smoothing, and quantification of signals from the sensor array. First, ultrasound measurements of a particular time period are excluded as noises if those of the microphone or IMU of the time period exceed the designated values (Figure 2a). Second, all the filtered ultrasound signals are converted to positive values and then smoothed by a simple moving average algorithm (Figure 2b). Lastly, the smoothed signals are quantified by six measures (1st peak amplitude, 2nd peak amplitude, peak-to-peak (PP) interval, duration, energy, and the number of peaks) referring to Lee [43], as shown in Figure 2c. Note that *peak* is operationally defined as the signal finishes increasing and starts decreasing while keeping the amplitude of more than 0.1 mV, *PP time interval* (unit: ms) as the time interval between the 1st and 2nd peaks, *duration* (unit: ms) as the time difference between the starting point when the signal begins to increase above a designated level and the ending point when the signal reaches the baseline level, and *energy* (unit: mV^2^) as the sum of the squared amplitudes during the duration [44].

## 3. Experiment of Swallowing

### 3.1. Participants

A total of 24 healthy participants (14 males and 10 females; age = 30.5 ± 7.6 years) were recruited to examine the characteristics of ultrasound signals by pharyngeal swallowing using the SMAS. Any history of swallowing disorders or problems with food intake were reported by the participants.

### 3.2. Experiment Procedure

The swallowing experiment was performed in the following four phases: introduction, preparation, measurement, and debriefing. In the introduction phase, the purpose and procedure of the experiment were explained to the participant and informed consent was obtained. In the preparation phase, the SMAS was placed around the neck, the ultrasonic Doppler sensor array coated with water-soluble gel was vertically attached with an elastic band near the right side of the laryngeal prominence, and then the participant was asked to say his/her name aloud to check if the signals of the ultrasonic Doppler sensor array were adequately measured. In the measurement phase, signals were recorded three times for each of four swallowing conditions: 3 mL and 9 mL of thin liquid (water) and thick liquid (Yoplait Plain, Binggrae Co., Ltd., Republic of Korea). The experiment was planned to immediately stop if aspiration occurred. Lastly, the SMAS was dismounted, a debriefing was conducted, and monetary compensation was provided for the participation. The experiment was approved (PIRB-2019-E021) by the Institutional Review Board at Pohang University of Science and Technology.

### 3.3. Analysis Methods

An ANOVA was conducted to identify the significance of bolus viscosity and bolus volume on the ultrasound signal measures (1st peak amplitude, 2nd peak amplitude, PP time interval, duration, and energy) of pharyngeal swallowing. Pearson’s correlation analysis was performed to examine the relationships between the ultrasound signal measures. Outliers were removed so that the coefficients of variation (CVs) of swallowing signals were kept below particular levels (CV < 0.4 for the amplitude and energy measures and CV < 0.2 for the time measures). A paired *t*-test was conducted for post-hoc analysis of significant factors and a z-test was conducted for testing the percentage of two or more peaks by bolus viscosity and bolus volume. Statistical testing in the present study was conducted at *α* = 0.05 unless otherwise specified and Minitab 19 (Minitab LLC., State College, PA) was used for statistical analysis.

## 4. Results

An ANOVA with two within-subjects factors (viscosity and volume) on swallowing ultrasound signal measurements (Table 1) shows that viscosity was found significant on 1st peak amplitude, 2nd peak amplitude, and energy and volume on 1st peak amplitude, PP time interval, and duration. No interaction effects between viscosity and volume were found significant for any of the swallowing measures.

### 4.1. Peak Amplitudes

As displayed in Figure 3, the average 1st peak amplitude changed more by viscosity (23.6%~29.0%) than by volume (8.2%~16.4%). Figure 3a shows that, as compared to the average 1st peak amplitude of thin liquid, that of thick liquid significantly decreased by 23.6% (0.13 mV) for 3 mL (*t*_[18]_ = 4.74, *p* < 0.01) and 29.0% (0.18 mV) for 9 mL (*t*_[18]_ = 7.47, *p* < 0.01). On the other hand, Figure 3b shows that, as compared to the average 1st peak amplitude of 3 mL, that of 9 mL increased by 16.4% (0.09 mV) with significance for thin liquid (*t*_[18]_ = −2.49, *p* = 0.02) and 8.2% (0.04 mV) without significance for thick liquid (*t*_[18]_ = −1.86, *p* = 0.08).

As displayed in Figure 4, the average 2nd peak amplitude changed greater by viscosity (9.2%) than by volume (4.4%). The average 2nd peak amplitude of thick liquid was found significantly lower (0.04 mV) than that of thin liquid (*t*_[37]_ = 2.24, *p* = 0.03). On the other hand, the average changes (0.02 mV) in 2nd peak amplitude by volume were not found significant (*t*_[37]_ = −1.01, *p* = 0.32).

### 4.2. Peak-to-Peak (PP) Time Interval

As displayed in Figure 5, the average PP time interval increased slightly more by volume (5.9%) than by viscosity (5.2%). The average changes (29.4 ms) in PP time interval by viscosity were not found significant (*t*_[37]_ = −1.56, *p* = 0.13). On the other hand, the average PP time interval of 9 mL was found significantly longer (33.2 ms) than that of 3 mL (*t*_[37]_ = −2.12, *p* = 0.04).

### 4.3. Duration

As displayed in Figure 6, the average duration increased greater by volume (5.6%) than by viscosity (1.4%). The average changes (12.5 ms) in duration by viscosity were not found significant (*t*_[37]_ = 0.60, *p* = 0.55). On the other hand, the average duration of 9 mL was found significantly longer (49.1 ms) than that of 3 mL (*t*_[37]_ = −2.79, *p* = 0.01).

### 4.4. Energy

As displayed in Figure 7, the average energy changed greater by viscosity (44.9%~47.6%) than by volume (17.0%~22.9%). Figure 7a shows that, as compared to the average energy of thin liquid, that of thick liquid significantly decreased by 44.9% (37.9 mV^2^) for 3 mL (*t*_[18]_ = 5.01, *p* < 0.01) and 47.6% (49.3 mV^2^) for 9 mL (*t*_[18]_ = 5.25, *p* < 0.01). On the other hand, Figure 7b shows that, as compared to the average energy of 3 mL, that of 9 mL increased by 22.9% (19.3 mV^2^) with significance for thin liquid (*t*_[18]_ = −4.04, *p* < 0.01) and 17.0% (7.9 mV^2^) without significance for thick liquid (*t*_[18]_ = −1.91, *p* = 0.07).

### 4.5. Correlations between Swallowing Signal Measures

As displayed in Figure 8, significant correlations were found for all the ten pairs of the five swallowing signal measures, except two pairs (1st peak amplitude and duration; 2nd peak amplitude and PP time interval). Of the significant correlations, strong correlations (*r* ≥ 0.7) were found for two pairs (1st peak amplitude and energy; PP time interval and duration) and moderate correlations (0.7 < *r* < 0.4) for one pair (2nd peak amplitude and energy). The correlation between the time measures (*r* = 0.78) was higher than the correlation between the peak amplitude measures (*r* = 0.30); those between the time measures and the peak amplitude measures were found low (*r* = −0.27 to 0.18). Lastly, energy was found correlated higher with the peak amplitude measures (*r* = 0.86 with 1st peak amplitude and *r* = 0.5 with 2nd peak amplitude) than with the time measures (*r* = −0.24 with PP time interval and *r* = 0.17 with duration).

### 4.6. Number of Peaks

As displayed in Figure 9, the proportion of two or more peaks changed more largely by viscosity (∆ = 11.1%) than by volume (∆ = 7.9%). Figure 9a shows that the proportion of two or more peaks of thin liquid (87.9%) was significantly higher than that of thick liquid (76.8%; *z* = 2.03, *p* = 0.04). Next, Figure 9b shows that the proportion of two or more peaks of 9 mL (86.5%) slightly increased without significance compared to that of 3 mL (78.6%; *z* = 1.46, *p* = 0.15).

### 4.7. Reproducibility of Measurement

The distribution of the coefficient of variation (CV, the ratio of the standard deviation to the mean) presented in Table 2 indicates that the reproducibility of the measurement varies mainly by ultrasonic measure and that duration has the highest reproducibility, followed by the PP time interval, energy, 1st peak amplitude, and 2nd peak amplitude. The CV of the three repeated measurements collected in each swallowing condition was calculated, and then the distribution of the CV was constructed by bolus viscosity, bolus volume, and ultrasonic measure, as displayed in Table 2. More than 80% of the measurements of duration and PP time interval were found at a CV ≤0.2 and those of energy, 1st peak amplitude, and 2nd peak amplitude were found at a CV ≤0.4. On the basis of the CV analysis results, the present study excluded those of duration and PP time interval if their CV was >0.2 and those of energy, 1st peak amplitude, and 2nd peak amplitude if their CV was >0.4.

## 5. Discussion

The present study develops a wireless neck-band-type SMAS comprising a slim and curved ultrasonic Doppler sensor array, a microphone, and an IMU so that ultrasound signals due to swallowing can be collected effectively. The single-crystal flat disk transducer (diameter = 28 mm and thickness = 33.5 mm) of a portable ultrasonic detector DF-4001 (Martec Med LLC., Brazil) used by Soria et al. [45], Cagliari et al. [31], and Santos and Filho [33] needs to be located carefully on the lateral tracheal border just below the cricoid cartilage during swallowing measurement while neck-related motions such as speech production, coughing, and/or movements other than swallowing are restricted. On the other hand, the SMAS has a sensor array consisting of two transmitters and five receivers with 133 mm of curvature and 5 mm of thickness so that the sensor array can be located easily on the neck, because the swallowing detection range of the SMAS is wider than that of the single-sensor of the ultrasonic detector DF-4001. Furthermore, the SMAS does not require restrictions of neck-related motions and the curved strip-type sensor array can be placed on the neck easily, securely, and with a snug fit, using an elastic band. The signal processing algorithm of the SMAS can exclude ultrasound measurements as noises from the subsequent analysis if measurements of the microphone or the IMU exceed a designated value by assuming that the corresponding measurements occur due to activities other than swallowing. To our best knowledge, the ultrasonic Doppler sensor-based system developed in the present study is the first of its kind that is wearable, wireless, and measures signals originating only from swallowing activities.

The results of the ANOVA and correlation analysis in the present study identified the effects of the viscosity and volume of bolus on the swallowing ultrasound signal measures (1st peak amplitude, 2nd peak amplitude, duration, PP time interval, energy, and number of peaks) and the relationships of the ultrasound signal measures. To our best knowledge, no previous studies have examined the effects of the viscosity and volume of bolus on the swallowing ultrasound signal measures and the relationships of the ultrasound signal measures. Note that Soria et al. [45], Cagliari et al. [31], and Santos and Filho [33] used an ultrasonic fetal detector reported ultrasound signal measurements for various bolus viscosity and volume conditions and tested their differences between age groups. The ANOVA result showed that viscosity was significant mainly on the amplitude and energy measures, volume was significant on the time measures, and the interaction between viscosity and volume was not significant on all the swallowing signal measures. Next, the correlation analysis results found the time measures strongly correlated (*r* = 0.78), the amplitude measures weakly correlated (*r* = 0.30), the amplitude measures crossed the time measures weakly correlated (*r* = −0.27 to 0.18), and energy having higher correlations with the amplitude measures (*r* = 0.86 with 1st peak amplitude and *r* = 0.5 with 2nd peak amplitude) than with the time measures (*r* = −0.24 with PP time interval and *r* = 0.17 with duration).

The present study identified that the 1st peak amplitude (23.6%~29.0%), 2nd peak amplitude (9.2%), energy (44.9%~47.6%), and proportion of two or more peaks (11.1%) significantly decreased as the viscosity increased. Note that the amplitude of the ultrasound signal decreases as the velocity of a moving object decreases [46]. Thus, the reductions in peak amplitudes, energy, and proportion of two or more peaks can be explained by the movements of the swallowing-related organs being slower for thick liquid with a high viscosity than for thin liquid with a low viscosity. A similar result was reported by Cagliari et al. [31]: a decrease in the peak intensity of the ultrasound signal as an increase in the viscosity of bolus (91.1~92.7 dB for water and 90.0~92.4 dB for yogurt). Lastly, the first and second peaks can be related to the elevation of the hyolaryngeal complex and its return to the original position, respectively, which needs to be substantiated by VFSS. The hyolaryngeal complex is elevated by the synergistic contraction of the suprahyoid, thyrohyoid, and long pharyngeal muscles (leading to the opening of the upper esophageal sphincter and transferring of a bolus from the oral cavity to the esophagus) and finally returns to its original position [47].

Next, the present study found that the 1st peak amplitude (8.2%~16.4%), PP time interval (5.9%), duration (5.6%), energy (17.0%~22.9%), and proportion of two or more peaks (7.9%) increased as the bolus volume increased to 9 mL from 3 mL. Note that the effects of the volume on the 1st peak amplitude and energy were significant only for thin liquid. The increase in the peak amplitude and energy can be interpreted as the faster movements of the swallowing-related organs caused by higher innervations commanded from the brain for 9 mL than those for 3 mL. Lastly, the increases in the PP time interval and duration by the bolus volume are natural phenomena.

As shown in Table 3, the average swallowing durations of the healthy participants (902.2 ± 149.3 ms for thin liquid and 889.7 ± 163.8 ms thick liquid) in the present study were found similar (difference <10%) to those reported by Cagliari et al. [31], who used an ultrasonic Doppler sensor, and Nascimento et al. [48], who used a videofluoroscopy, but quite different (difference >30%) from those reported by Santamato et al. [32] and Youmans and Stierwalt [49], who used a microphone. It is noteworthy that shorter durations for thick liquid than those for thin liquid were commonly found in the ultrasonic Doppler sensor-based studies, while the opposite was found in the videofluoroscopy-based study. Note that increases in the swallowing duration by the bolus volume were commonly observed in previous studies. Since the swallowing analysis results using an ultrasonic Doppler sensor are more similar with those using the gold standard videofluoroscopy than with those using a microphone, it can be inferred that the ultrasonic Doppler sensor technique has higher validity than the microphone technique in swallowing research.

The wireless neck-band-type SMAS in the study can be effectively used to monitor swallowing activities in daily life, quantify swallowing functions, and screen those with swallowing problems. As VFSS and FEES are limited in terms of intrusiveness and subjectivity, the ultrasonic Doppler sensor-based SMAS can monitor swallowing activities safely and unobtrusively in daily life. Quantitative information such as peak amplitude, PP time interval, duration, energy, and the number of peaks of ultrasound signals from swallowing can be analyzed and their changes before and after particular treatments are reported. Lastly, screening services for dysphagia are possible using the SMAS by monitoring swallowing activities and identifying an abnormal pattern of swallowing.

Further research is needed to examine the concurrent validity and generalizability of the ultrasonic Doppler sensor-based SMAS. A comparative analysis of ultrasound signals and VFSS images is needed to check the concurrent validity of the SMAS. Although the trend of SMAS signals measured from the healthy was examined in the present study, the significance of the swallowing ultrasound signal measures needs to be understood by relating to VFSS analysis results. Furthermore, the measurements of the SMAS need to be collected from those diagnosed with dysphagia at various severity levels by various causes including neurological disorders, congenital conditions, and muscular conditions and then compared with the healthy. Then, a statistical model for the screening of dysphagia using the SMAS can be developed by gathering large-scale normative data of the healthy and patients with dysphagia.

## Figures and Tables

**Figure 1 sensors-20-04529-f001:**
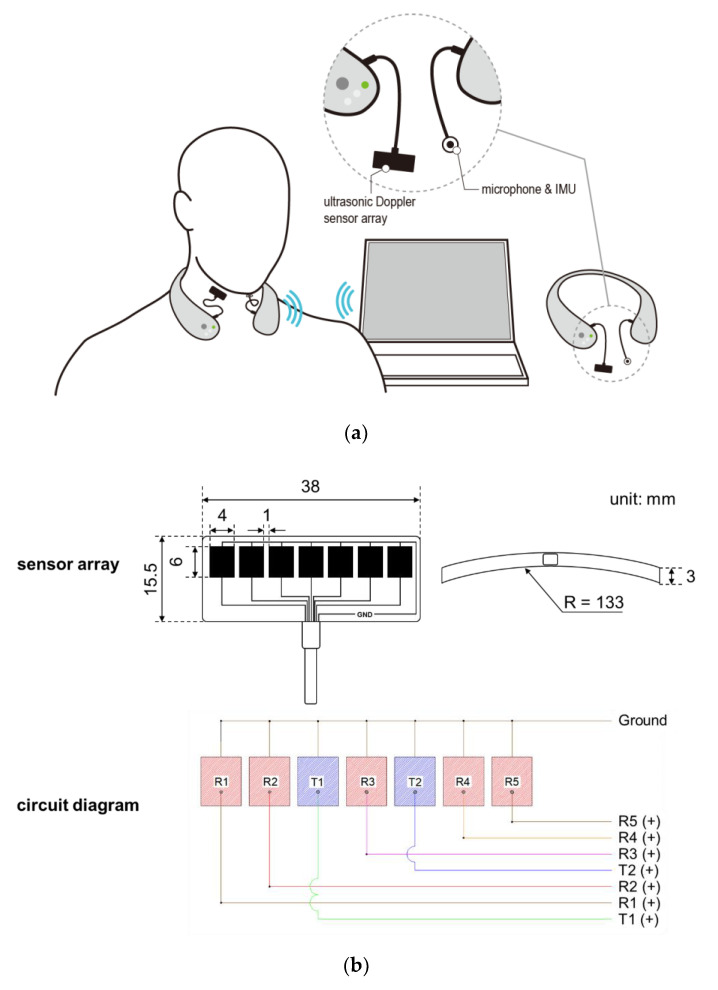
Wireless swallowing monitoring system: (**a**) system configuration; (**b**) five-ultrasonic Doppler sensor array consisting of two transmitters and five receivers.

**Figure 2 sensors-20-04529-f002:**
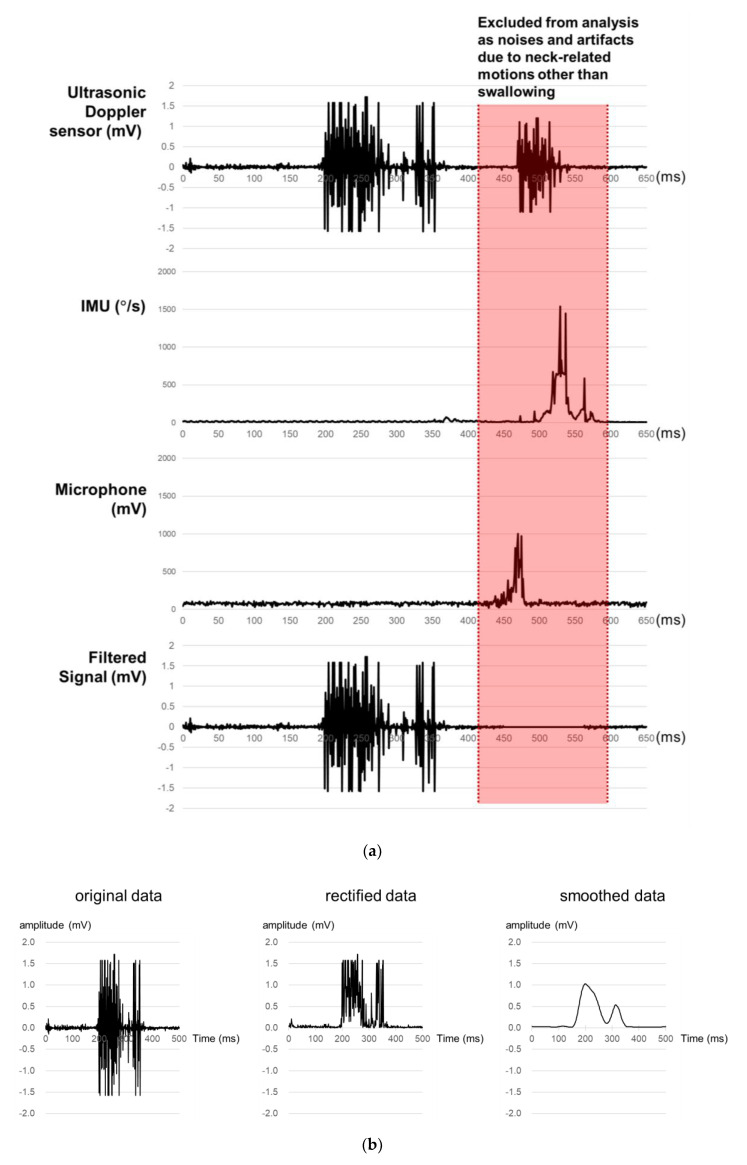
Signal processing for ultrasound signals for swallowing analysis: (**a**) noise removal; (**b**) signal rectification and smoothing; (**c**) signal quantification.

**Figure 3 sensors-20-04529-f003:**
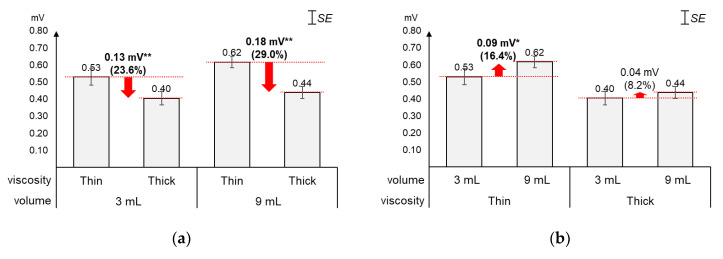
Average 1st peak amplitudes: (**a**) by bolus viscosity and bolus volume; (**b**) by bolus volume and bolus viscosity (*: *p* < 0.05; **: *p* < 0.01).

**Figure 4 sensors-20-04529-f004:**
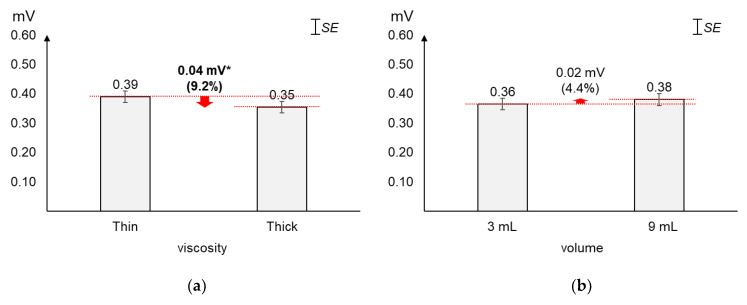
Average 2nd peak amplitudes: (**a**) by bolus viscosity; (**b**) by bolus volume (*: *p* < 0.05; **: *p* < 0.01).

**Figure 5 sensors-20-04529-f005:**
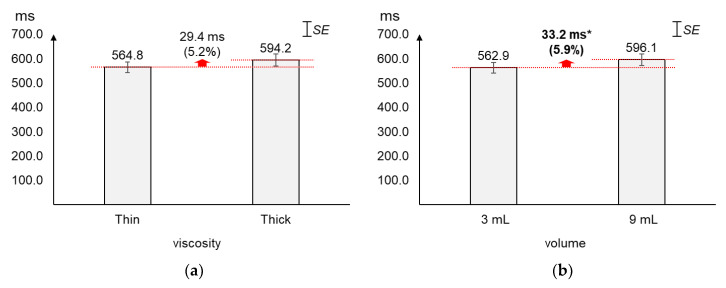
Average peak-to-peak time intervals: (**a**) by bolus viscosity; (**b**) by bolus volume (*: *p* < 0.05; **: *p* < 0.01).

**Figure 6 sensors-20-04529-f006:**
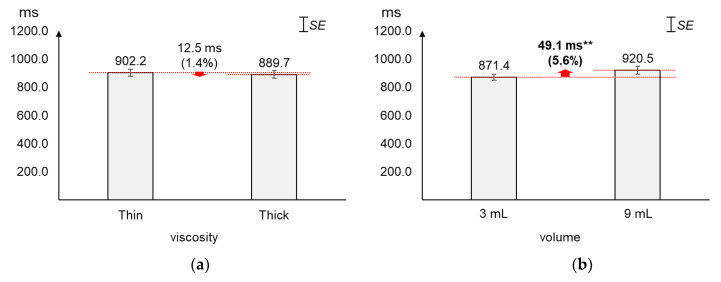
Average duration: (**a**) by bolus viscosity; (**b**) by bolus volume (*: *p* < 0.05; **: *p* < 0.01).

**Figure 7 sensors-20-04529-f007:**
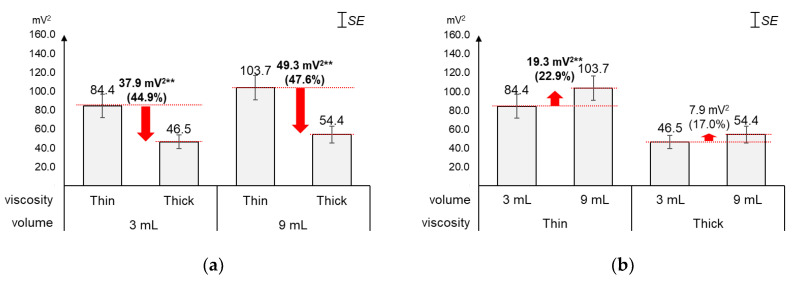
Average energies: (**a**) by bolus viscosity and bolus volume; (**b**) by bolus volume and bolus viscosity (*: *p* < 0.05; **: *p* < 0.01).

**Figure 8 sensors-20-04529-f008:**
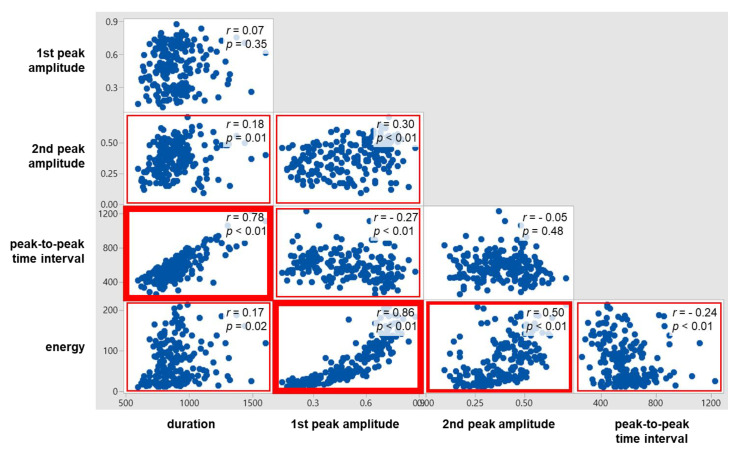
Correlations between swallowing ultrasound signal measures.

**Figure 9 sensors-20-04529-f009:**
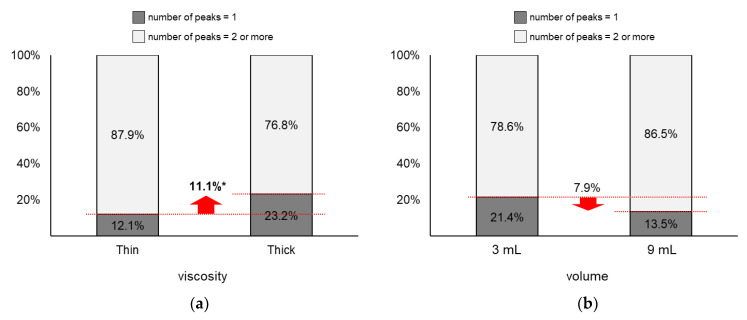
Proportions of two or more peaks: (**a**) by bolus viscosity; (**b**) by bolus volume (*: *p* < 0.05; **: *p* < 0.01).

**Table 1 sensors-20-04529-t001:** ANOVA results of swallowing ultrasound signal measurements (*: *p* < 0.05; **: *p* < 0.01).

Measure	Source	*df*	SS	MS	*F*	
1st peak amplitude (mV)	Viscosity [Vc]	1	1.080	1.080	80.8	**
	Vc × S	18	0.243	0.013		
	Volume [Vl]	1	0.168	0.168	9.0	**
	Vl × S	18	0.339	0.019		
	Vc × Vl	1	0.034	0.034	2.0	
	Vc × Vl × S	18	0.300	0.017		
	Subject [S]	18	4.257			
	Total	193	6.963	1.572		
2nd peak amplitude (mV)	Viscosity [Vc]	1	0.060	0.060	4.6	*
	Vc × S	18	0.233	0.013		
	Volume [Vl]	1	0.012	0.012	0.9	
	Vl × S	18	0.230	0.013		
	Vc × Vl	1	0.005	0.005	0.5	
	Vc × Vl × S	18	0.195	0.011		
	Subject [S]	18	1.962			
	Total	193	3.247	0.226		
peak-to-peak (PP) time interval (ms)	Viscosity [Vc]	1	40,104	40,104	1.8	
	Vc × S	18	414,691	23,038		
	Volume [Vl]	1	51,593	51,593	4.5	*
	Vl × S	18	206,003	11,445		
	Vc × Vl	1	14,165	14,165	1.2	
	Vc × Vl × S	18	209,110	11,617		
	Total	193	4,646,565	322,590		
duration (ms)	Viscosity [Vc]	1	7217	7217	0.2	
	Vc × S	18	539,724	29,985		
	Volume [Vl]	1	112,522	112,522	5.8	*
	Vl × S	18	352,796	19,600		
	Vc × Vl	1	5872	5,872	0.5	
	Vc × Vl × S	18	199,791	11,099		
	Subject [S]	18	3,375,375			
	Total	193	5,267,587	379,218		
energy	Viscosity [Vc]	1	88,735	88,735.1	30.3	**
	Vc × S	18	53,254	2958.6		
	Volume [Vl]	1	8639	8639.5	21.5	**
	Vl × S	18	7242	402.3		
	Vc × Vl	1	1513	1512.7	2.9	
	Vc × Vl × S	18	9430	523.9		
	Subject [S]	18	324,209			
	Error	118	38,077	322.7		
	Total	193	540,733	121,106.4		

**Table 2 sensors-20-04529-t002:** Distribution of coefficient of variation of swallowing ultrasound signal measurements (*n* = 24)

Viscosity	Volume	Measures	Coefficient Of Variation					
			≤0.1	0.1–0.2	0.2–0.3	0.3–0.4	0.4–0.5	>0.5
Thin	3 mL	1st Peak amplitude	33.3%	33.3%	16.7%	8.3%	8.3%	0.0%
		2nd Peak amplitude	20.8%	25.0%	16.7%	12.5%	16.7%	8.3%
		PP time interval	62.5%	20.8%	8.3%	4.2%	0.0%	4.2%
		Duration	79.2%	16.7%	0.0%	4.2%	0.0%	0.0%
		Energy	12.5%	25.0%	29.2%	20.8%	0.0%	12.5%
	9 mL	1st Peak amplitude	41.7%	33.3%	12.5%	4.2%	8.3%	0.0%
		2nd Peak amplitude	16.7%	25.0%	25.0%	16.7%	0.0%	16.7%
		PP time interval	54.2%	29.2%	8.3%	0.0%	0.0%	8.3%
		Duration	79.2%	12.5%	0.0%	8.3%	0.0%	0.0%
		Energy	20.8%	25.0%	12.5%	29.2%	4.2%	8.3%
Thick	3 mL	1st Peak amplitude	29.2%	45.8%	25.0%	0.0%	0.0%	0.0%
		2nd Peak amplitude	12.5%	33.3%	25.0%	0.0%	12.5%	16.7%
		PP time interval	41.7%	29.2%	20.8%	4.2%	4.2%	0.0%
		Duration	70.8%	12.5%	16.7%	0.0%	0.0%	0.0%
		Energy	4.2%	37.5%	25.0%	12.5%	12.5%	8.3%
	9 mL	1st Peak amplitude	20.8%	37.5%	29.2%	12.5%	0.0%	0.0%
		2nd Peak amplitude	12.5%	12.5%	41.7%	16.7%	8.3%	8.3%
		PP time interval	37.5%	50.0%	4.2%	8.3%	0.0%	0.0%
		Duration	75.0%	16.7%	8.3%	0.0%	0.0%	0.0%
		Energy	8.3%	16.7%	20.8%	33.3%	16.7%	4.2%

**Table 3 sensors-20-04529-t003:** Comparison of swallowing duration results of healthy participants

Study	Sample Size (Age Range, Years)	Measurement Device	Bolus (Volume)	Duration (ms)	
				Mean	SD
Present study	24 (23~49)	ultrasonic Doppler	water (3 mL, 9mL)	902.2	149.3
			yogurt (3mL, 9mL)	889.7	163.8
Cagliari et al. (2009)	30 (10~15)	ultrasonic Doppler	water (2.5 mL)	male: 990.0 female: 970.0	N/A *
			yogurt (2.5 mL)	male: 920.0 female: 810.0	N/A
Nascimento et al. (2015)	30 (29~77)	videofluoroscopy	water + barium (5 mL, 10 mL)	832.5	N/A
			honey + barium (5 mL, 10 mL)	936.0	N/A
Santamato et al. (2009)	60 (>18)	microphone	water (10 mL)	438.1	109.6
			yogurt (10 mL)	564.2	168.2
Youmans & Stierwalt (2005)	97 (20~79)	microphone	water (5 mL)	490.0	130.0
			honey (5 mL)	550.0	110.0

* N/A: not available.

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
