# Peer review of "Development of an Ultrasonic Doppler Sensor-Based Swallowing Monitoring and Assessment System"

_sensors, 2020, doi:10.3390/s20164529_

Round 1

Reviewer 1 Report

Dear authors:

Even though the manuscript has interesting results, the English is far to be appropriate for publication, terms like measure (s) (instead of measurements), duration time (look redundant), and many others should be revised in the entire manuscript. I suggest to polish the English with the help of a professional translator. 

I have some doubts about the novelty of this work because Cagliari, C. F.; Jurkiewicz, A. L.; et al. published something similar years before, acoording to your references. What is new in your work?

Author Response

  1. Even though the manuscript has interesting results, the English is far to be appropriate for publication, terms like measure(s) (instead of measurements), duration time (look redundant), and many others should be revised in the entire manuscript. I suggest to polish the English with the help of a professional translator.

⇒ The manuscript has been proofread and revised with the help of two native English speakers providing professional editorial services.

  1. I have some doubts about the novelty of this work because Cagliari, C. F.; Jurkiewicz, A. L.; et al. published something similar years before, according to your references. What is new in your work?

⇒ Compared to the existing swallowing studies using an ultrasound Doppler sensor, the novelty of our is two-fold as follows:

  • The present study developed an ultrasound Doppler sensor-based system which is wearable, wireless, and measuring signals originated only from swallowing activities. In contrast, the existing studies used a portable fetal detector which requires a careful placement of the transducer on the neck as well as a restriction of neck motions during swallowing measurement.
  • The present study examined the effects of bolus viscosity and bolus volume on swallowing ultrasound signal measures and the relationships of the ultrasound signal measures, while the existing studies simply reported swallowing ultrasound signal measurements and compared them between age groups.

The novelty of our work has been highlighted in the manuscript by accommodating the comment of the reviewer.

Reviewer 2 Report

Existing swallowing evaluation methods using X-ray or endoscopy are qualitative. The present study developed a swallowing monitoring and assessment system (SMAS) that is nonintrusive and quantitative. As a results, authors suggested that the proportion of two or more peaks varied from 76.8% to 87.9% by viscosity and volume. Further research is needed to examine the concurrent validity and generalizability of the ultrasonic Doppler sensor-based SMAS.

I know understand this manuscript's importance.

However, it need to revised a few points.

1) I did not know that this manuscript's hypothesis.

So could you decrive this exactly?

2) Is this method reproducible?

3) A more detailed description of the figures description are required. These are hard to understand

Author Response

  1. I did not know that this manuscript's hypothesis. So could you describe this exactly?

⇒ The present study is an exploratory study which does not require any prior assumptions or hypotheses. As described in the manuscript, the present study was conducted to examine the effects of bolus viscosity and bolus volume on swallowing ultrasound signal measures and the relationships between the ultrasound signal measures. Necessary expressions have been added in the Introduction section to clarify the research type of the present study.

  1. Is this method reproducible?

⇒ The reproducibility analysis results of our measurement method of pharyngeal swallowing function using an ultrasonic Doppler sensor array have been added in the Results section. The reproducibility of our proposed method cannot be compared with those of the existing studies due to unavailability of the corresponding information.

  1. More detailed descriptions of the figures are required. These are hard to understand

⇒ The labels, captions, and descriptions of the figures have been added for better understanding of the figures.

Round 2

Reviewer 1 Report

The manuscript is ready for its publication 

Reviewer 2 Report

This manuscript was revised based on the reviewers comments.

So, I recommend to publish this.